# Preoperative Predictors of Adverse Clinical Outcome in Emergent Repair of Acute Type A Aortic Dissection in 15 Year Follow Up

**DOI:** 10.3390/jcm10225370

**Published:** 2021-11-18

**Authors:** Miriam Freundt, Philipp Kolat, Christine Friedrich, Mohamed Salem, Matthias Gruenewald, Gunnar Elke, Thomas Pühler, Jochen Cremer, Assad Haneya

**Affiliations:** 1Department of Cardiovascular Surgery, University Medical Center Schleswig-Holstein, Campus Kiel, Arnold-Heller-Straße 3, 24105 Kiel, Germany; Philipp.Kolat@uksh.de (P.K.); Christine.Friedrich@uksh.de (C.F.); Mohamed.Salem@uksh.de (M.S.); Thomas.Puehler@uksh.de (T.P.); Jochen.Cremer@uksh.de (J.C.); assad.haneya@uksh.de (A.H.); 2Heart and Vascular Institute, Intensive Care Unit, Penn State Health Milton S. Hershey Medical Center, Hershey, PA 23538, USA; 3Department of Anaesthesiology and Intensive Care Medicine, University Medical Center Schleswig-Holstein, Campus Kiel, Arnold-Heller-Str. 3 Haus R3, 24105 Kiel, Germany; Matthias.Gruenewald@uksh.de (M.G.); Gunnar.Elke@uksh.de (G.E.)

**Keywords:** predictor, adverse outcome, emergent surgical repair, acute type A dissection

## Abstract

Background: Acute type A aortic dissection (AAAD) has high mortality. Improvements in surgical technique have lowered mortality but postoperative functional status and decreased quality of life due to debilitating deficits remain of concern. Our study aims to identify preoperative conditions predictive of undesirable outcome to help guide perioperative management. Methods: We performed retrospective analysis of 394 cases of AAAD who underwent repair in our institution between 2001 and 2018. A combined endpoint of parameters was defined as (1) 30-day versus hospital mortality, (2) new neurological deficit, (3) new acute renal insufficiency requiring postoperative renal replacement, and (4) prolonged mechanical ventilation with need for tracheostomy. Results: Total survival/ follow-up time averaged 3.2 years with follow-up completeness of 94%. Endpoint was reached by 52.8%. Those had higher EuroSCORE II (7.5 versus 5.5), higher incidence of coronary artery disease (CAD) (9.2% versus 3.2%), neurological deficit (ND) upon presentation (26.4% versus 11.8%), cardiopulmonary resuscitation (CPR) (14.4% versus 1.6%) and intubation (RF) before surgery (16.9% versus 4.8%). 7-day mortality was 21.6% versus 0%. Hospital mortality 30.8% versus 0%. Conclusions: This 15-year follow up shows, that unfavorable postoperative clinical outcome is related to ND, CAD, CPR and RF on arrival.

## 1. Introduction

Acute type A aortic dissection (AAAD) is a catastrophic event in which the inner layer of the ascending aorta tears and separates from the middle layer. Blood surges into the false lumen, which can result in multiple organ damage due to hypoperfusion. The condition can quickly deteriorate into shock, hemodynamic instability and death. Emergent surgical repair remains the gold standard of care. Due to acuity of the illness preoperative evaluation is limited, immediate decisions have to be made by surgeons and postoperative adverse clinical outcome remains oftentimes of concern [1,2,3,4,5,6,7,8]. Without treatment mortality increases dramatically by the hour and has been reported as high as 1 to 3% per hour during the first 24 h, 30% after one week, 80% after two weeks, and 90% at one year [9]. About 20% of patients with AAAD die before even reaching the hospital [9]. A recent multi-institutional study across all emergency rooms in Berlin, Germany from 2006 to 2016 showed an incidence of AAAD of 5.24 cases in 100,000 visits per year but based on the city’s autopsy results 50% of AAAD had remained undetected [10]. Even with surgical repair mortality is high and ranges up to 16–27% within 30-days [11,12,13]. We may be able to reduce mortality with advances in surgical strategies and perioperative critical care, but functional status and quality of life (QoL) in survivors are becoming an increasing concern since simply surviving surgery but then ending up in an overall devastating condition must not be a goal. Alterations in lifestyle and emotional state are common in survivors of AAAD and many patients are unable to return to their previous occupation [14]. Previous studies investigating the survival of AAAD patients have been published. But there is only scarce data on the effect of preoperative risk factors on clinical outcome of these patients. Hence, the aim of this study was to associate obvious preoperative conditions with a combined endpoint of undesirable adverse clinical outcome, that might guide clinicians in future decision-making.

## 2. Materials and Methods

### 2.1. Study Design and Patient Population

We performed a retrospective analysis of our Aortic Dissection Register, which included all consecutive 394 cases of AAAD who underwent emergent repair in moderate hypothermic cardiac arrest (MHCA) in our institution between 2001 and 2018. AAAD was defined as dissection of the aortic wall that involved the ascending aorta with extension to the arch or descending aorta, regardless of the site of the primary intimal tear. Variants with aortic intramural hematoma and intimal tears without hematoma as well as penetrating atherosclerotic ulcers were included. Diagnosis was generally established with emergent computed tomographic (CT) angiography of the chest, abdomen, and pelvis. Bedside transthoracic echocardiography was used to assess the presence of pericardial effusion and overall left ventricular function and in addition patients routinely underwent transesophageal echocardiography after induction of general anesthesia and endotracheal intubation in the operating room to evaluate heart valves for need for concomitant procedures. A combined endpoint of four clinical outcome parameters was defined as (1) 30-day versus hospital mortality, (2) new neurological deficit, (3) new acute renal insufficiency requiring postoperative renal replacement therapy, and (4) prolonged mechanical ventilation with need for tracheostomy. Follow-up was conducted in May 2020 and long-term survival was evaluated by information given by the registry office.

### 2.2. Operative Technique and Postoperative Management

All cases were performed by experienced senior surgeons under general anesthesia in supine position with standard hemodynamic monitoring. All patients underwent median sternotomy and longitudinal pericardiotomy with cardiopulmonary bypass (CPB) in MHCA. The temperature probe was positioned in the nasopharynx and goal temperature was kept between 20 to 24 °C. From 2001 to 2010 arterial cannulation was achieved either by echocardiogram guided direct cannulation of the distal ascending aorta, the aortic arch, the apex, or either through the femoral or subclavian artery after surgical cut down. Starting in 2010 we gradually changed our standard approach for arterial cannulation to trans-atrial cannulation of the left ventricle via the right upper pulmonary vein [15]. The standard approach for venous drainage was cavoatrial cannulation with a common two-stage single venous cannula. Alternatively, we used echo guided cannulation of the femoral vein with a cannula extending into the right atrium or bicaval cannulation. Generally, we used retrograde injection of cold blood cardioplegic solution for myocardial protection after cross-clamping of the aorta. Bilateral antegrade cerebral perfusion with oxygenated cold blood (18 °C) was introduced through a balloon catheter inserted into the arch vessels with controlled flow pressure of 50–60 mmHg.

The origin and extend of the intimal tear determined the need for supracoronary ascending aortic replacement, partial versus total arch replacement with reimplantation of head and neck arteries, frozen elephant trunk, need for associated coronary artery bypass grafting or Conduit/Bentall procedure with reimplantation of coronary arteries versus David operation. After suturing of the distal anastomosis, the perfusion cannula was directly inserted into the graft. The aortic air was removed by resuming retrograde perfusion via the venous cannula followed by slow antegrade perfusion and then CPB was restarted. Continuous CO_2_ insufflation was used in addition. After the establishment of the proximal anastomosis, transesophageal echocardiography was done to rule out remaining intracardiac air. After primary hemostasis was achieved, the chest was closed, and the patient was brought to the cardiac intensive care unit (ICU) for standard postoperative care.

Patients were assessed for neurological deficit routinely every hour while in the ICU and every eight hours after transfer to the floor. In case of a new deficit, CT head was performed followed by formal neurological evaluation and magnetic resonance imaging of the brain to confirm the diagnosis. Kidney function was assessed every hour while in the ICU and every eight hours on the floor. In case of acute renal insufficiency renal replacement therapy was initiated after evaluation by a nephrologist or in case of severe electrolyte disturbances emergently. Mechanical ventilation was weaned per standard postoperative protocol with a goal for liberation as soon as possible. Tracheostomy was performed if weaning from mechanical ventilation and extubation was not possible within 10–12 days postoperatively.

### 2.3. Statistics

Normally distributed continuous variables were presented as mean ± standard deviation and compared by unpaired *t*-test. Categorical data were summarized as absolute (n) and relative (%) frequencies and compared by Chi^2^-test or Fisher’s exact test. Pre- and intraoperative variables were assessed for association with the combined endpoint by univariate analysis. 15-year survival was estimated by Kaplan-Meier curves. All tests were conducted 2-sided and a *p*-value of ≤0.05 was considered statistically significant. Data were analyzed with IBM SPSS Statistics for Windows (Version 24.0).

## 3. Results

Total survival/ follow-up time averaged 3.2 years with follow-up completeness of 94%. Follow-up was significantly shorter in the group who reached the combined endpoint, with 2.1 years versus 4.3 years, *p* < 0.001.

### 3.1. Preoperative Characteristics

The combined endpoint was reached by 52.8%. Patients who reached the endpoint had a significantly higher EuroSCORE II (7.5 versus 5.5, *p* < 0.001), higher incidence of coronary artery disease with previous percutaneous intervention (9.2% versus 3.2%, *p* = 0.016), higher incidence of neurological deficit upon presentation (26.4% versus 11.8%, *p* < 0.001), higher incidence of preoperative cardiopulmonary resuscitation (14.4% versus 1.6%, *p* < 0.001) and higher incidence of intubation before surgery (16.9% versus 4.8%, *p* < 0.001). There were no further significant differences with regard to clinical presentations between the groups. Table 1 shows detailed demographic and clinical characteristics of the study population.

### 3.2. Intraoperative Characteristics

Intraoperative characteristics are shown in Table 2. Patients who reached the combined endpoint also had significantly longer surgery duration (288 versus 256 min, *p* = 0.001), longer cardiopulmonary bypass times (180 versus 159 min, *p* < 0.001), longer cross-clamp time (96 versus 84 min, *p* = 0.010), and longer circulatory arrest (39 versus 32 min, *p* < 0.001). The requirement for intraoperative transfusion of blood products was higher in the group who reached the combined endpoint (number of units of red blood cells 4 versus 2, *p* < 0.001, number of units of fresh frozen plasma 1.5 versus 0, *p* = 0.031, number of pools of platelets 2 (ranging from 5 to 0) versus 2 (ranging from 4 to 0), *p* = 0.002). The need for total arch replacement was significantly higher in the group who reached the endpoint (21.2% versus 8.1%, *p* < 0.001). There were no differences between groups for all other surgical procedures such as single supracoronary replacement of the ascending aorta, partial arch replacement, Bentall operation, David Operation, Elephant trunk, associated coronary artery bypass grafting or cannulation site.

### 3.3. Postoperative Data and Outcome

Postoperative data and outcomes are shown in Table 3. Mortality was higher and complications were more common in the group who reached the combined end point. 7-day mortality was 21.6% versus 0%, *p* < 0.001. Hospital mortality was 30.8% versus 0%, *p* < 0.001. Causes of death were cardiac 53%, multiple organ failure in 43%, cerebral 9%, and sepsis 3%. The group who reached the endpoint had a significantly longer stay in the intensive care unit (10 days versus 4 days, *p* < 0.001), larger amount of postoperative drainage loss (1030 mL versus 750 mL, *p* < 0.001, greater need for postoperative blood transfusions (83.7% versus 64.5% of patients, *p* < 0.001), fresh frozen plasma transfusions (60.1% versus 40.3%, *p* < 0.001) and platelet transfusions (55.9% versus 37.6%, *p* < 0.001), as well as higher incidence of re-thoracotomy (26.9% versus 8.1%, *p* < 0.001). They also had a greater need for postoperative balloon pump and/or extracorporeal life support (5.1% versus 0.5%, *p* = 0.008), reintubation (27.9% versus 5.9%, *p* < 0.001), prolonged mechanical ventilation (189 h versus 24 h, *p* < 0.001) with need for tracheostomy (47.6% versus 0%, *p* < 0.001), readmission to the intensive care unit (13.5% versus 4.3%, *p* = 0.002), bacteremia/sepsis (8.7% versus 0.5%, *p* < 0.001), bronchopulmonary infection (22.1% versus 6.5%, *p* < 0.001), cardiac arrest (11.1% versus 2.2%, *p* < 0.001), new neurological deficit consistent with TIA/stroke (45.2% versus 0%, *p* < 0.001), myocardial infarction (2.9% versus 0%, *p* = 0.032), and acute renal insufficiency with need for renal replacement therapy (41.3% versus 0%, *p* < 0.001). While several parameters were less common in the group that reached the endpoint, they showed no statistical significance. Those were postoperative delirium (15.9% versus 21.1%), sternal wound infections (1.0% versus 2.2%) and atrial fibrillation (10.7% versus 10.2%).

### 3.4. Risk Factors for Combined Endpoint

Independent preoperative risk factors to reach the combined endpoint of mortality, new neurological deficit, prolonged mechanical ventilation with need for tracheostomy and acute renal insufficiency with need for renal replacement therapy were assessed with multivariable logistic regression analysis as shown in Table 4. Significant were coronary heart disease (*p* = 0.021, OR 2.122, CI 1.1–4.0), presence of a neurological deficit (*p* < 0.001, OR 3.6, CI 1.98–6.5), preoperative need for cardiopulmonary resuscitation (*p* = 0.001, OR 8.99, CI 2.5–32.3) and need for intubation on admission (*p* = 0.033, OR 2.5, CI 1.1–5.9).

### 3.5. Survival Curve

Figure 1 shows the Kaplan-Meier survival curve of patients who did and did not reach the combined endpoint with a follow-up time of 15 years. The group who reached the endpoint had significantly decreased 15-year survival, however, it is notable that curves are almost parallel, after the first 30-days, indicating that the highest rate of death occurs in the immediate postoperative period.

## 4. Discussion

It seems remarkable that the majority of patients who had complications did not just have one but multiple. Taken all facts into account, 52.8% of the patients in our population had an undesirable outcome.

Many previous studies have already evaluated risk factors for postoperative survival [9,16,17,18,19], but the universal ethical question remains in which high risk cases withholding surgery would provide less harm than performing it, since over 50% of survivors may have to tolerate devastating conditions on long term ventilation with tracheostomy, long-term dialysis and a severe neurological deficit. IRAD data indicated a mortality of 58% among those not receiving surgery, typically because of advanced age and comorbidity [20].

The intention of this study was to assess if undesirable post-operative outcome was associated with certain parameters present on presentation, to help eventually develop a strategy to know for which patient surgery is likely harmful. Accordingly, we chose a combination of severe debilitating complications as endpoint.

A common assumption is that patients with multiple underlying medical conditions such as hypertension, diabetes, hyperlipidemia, chronic kidney disease, smoking or previous cardiac surgery have adverse outcome [2]. However, in our study, the group who reached the combined endpoint had no higher incidence of such diagnoses, despite a higher EuroSCORE II. Therefore, the proposal is, that risk factors for developing aortic dissection are not applicable for suffering poor post-operative outcome. According to data from the Swedish National Diabetes Register patients with type 2 diabetes actually had significantly less risk of aortic aneurysm, dissection and reduced mortality after hospitalization compared to matched healthy controls. The authors hypothesized that glycated cross-links in aortic tissue may play a protective role in the progression of aortic diseases [21]. The previously published analysis of our database suggested that mortality was multifactorial and especially age, previous cardiac surgery, preoperative cardiopulmonary resuscitation, blood transfusion, and postoperative renal failure were considered risk factors [6].

According to our current data, the clinical condition in which the patient arrives preoperatively is predictive of poor outcome. Other authors showed as well that in-hospital adverse outcome was associated with the presence of lower limb hypoperfusion symptoms prior to surgery [2]. Since time is such an essential part, prompt diagnosis and referral to immediate surgical repair remain the main goal. Michael DeBakey once stated: “no physician can diagnose a condition he never thinks about”. An analysis from the International Registry of Acute Aortic Dissection (IRAD) indicates that the median time from emergency department presentation to definitive diagnosis of acute aortic dissection is 4.3 h, with an additional 4 h between diagnosis and surgical intervention for type A patients [22].

The response time for emergency medical services is legally regulated in Germany and should not exceed 12 min from alarm to arrival in our federal state [23], but even in densely populated areas averages 8–10 min. Emergency physicians ride on the ambulance and can make an immediate assessment. If AAAD is suspected, the physician alarms the emergency room personnel to have imaging available immediately on arrival, as well as the cardiovascular surgeon on stand-by. Despite these seemingly ideal conditions, analysis of the German Registry for Acute Aortic Dissection Type A including 2137 patients by Boening et al., revealed an overall 30-day mortality of 16.9% and new neurologic dysfunction postoperatively in 9.5% [11]. While our mortality coincides well with the national level, our rate of neurological complications seems to be higher ranging up to 23.9%. In another single center retrospective analysis by Haldenwang et al., the 30-day mortality rate was 16.4%. In their population 33.6% suffered transient neurological dysfunction and 16.4% had a postoperative stroke [5]. They also looked at a combined adverse outcome defined as stroke and 30-day mortality and found high body mass index, preoperative hypoperfusion syndrome, and left ventricular ejection fraction <50% to be independent predictors. Our results indicate a higher incidence of cardiopulmonary resuscitation within 48 h before surgery and preoperative mechanical ventilation in the combined endpoint group, but there was no higher incidence in the presence of IABP/ ECLS, cardiogenic shock, pericardial tamponade or decompensated renal insufficiency. There was also no higher prevalence of Marfan Syndrome or difference in average left ventricular ejection fraction.

Current risk assessment scores don’t seem to provide an accurate answer in AAAD, especially EuroSCORE II appears to underestimate mortality. Our current analysis does not evaluate postoperative QoL in such circumstances. A previous investigation within our group found however, that the QoL scores were lower one year after emergent surgery for AAAD compared to the general, age-matched population in Germany especially regarding pain score and social functioning [3].

With growing socioeconomic and financial pressure in hospitals and healthcare systems, early identification of patients at risk for prolonged length of hospital stay with needs for advanced therapies is also essential. It was no surprise that patients who reached the combined endpoint had significantly longer stays on the ventilator, in the intensive care unit as well as in the hospital compared to those who did not reach the endpoint.

Our results stress again the importance of early diagnosis of AAAD and immediate referral to a facility capable to operate immediately, since the clinical condition on arrival plays such an important role as prognostic marker.

This study is designed as single-center retrospective review of an internal database and not a randomized prospective trial. Information was obtained from our institutional database. Data were entered by staff physicians during the patients’ hospitalizations. Therefore, data may be subject to bias. From our data it remains unclear if and how our change of strategy regarding atrial canulation may have influenced the outcome.

## 5. Conclusions

We showed, in 15-year follow up, that relevant risk factors for adverse postoperative clinical outcome are rather related to the clinical condition in which the patient arrives preoperatively, than preexisting medical illnesses widely assumed to be responsible for poor outcome. This supports prioritizing immediate surgical attention to patients, before they may otherwise progress to hemodynamic instability and hypoperfusion even if they have underlying medical conditions or advanced age. The ethical dilemma arises when patients arrive at the hospital with already existing hypoperfusion, ongoing cardiopulmonary resuscitation or even intubation. In those cases, our data suggest that physicians may recommend either non-surgical treatment due to extremely poor chances for acceptable outcome or have a detailed discussion with patient and families of what to expect.

## Figures and Tables

**Figure 1 jcm-10-05370-f001:**
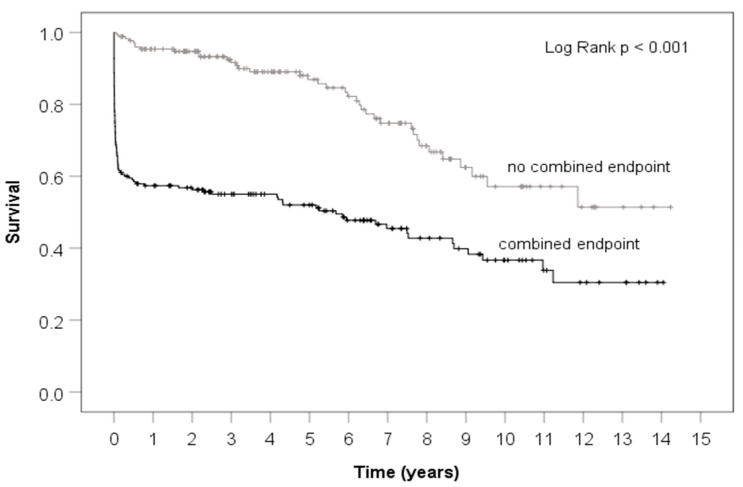
Survival curves of patients with and without reaching the combined endpoint.

**Table 1 jcm-10-05370-t001:** Demographic and clinical characteristics of the study population.

KERRYPNX	All Patients (*n* = 394)	Combined Endpoint = 0 (*n* = 186/47.2%)	Combined Endpoint = 1 (*n* = 208/52.8%)	*p*-Value
Age, years	62.5 ± 13.0	61.7 ± 14.0	63.2 ± 11.9	0.567
	63.0 (53.0;73.0)	63.0 (53.0;71.3)	63.5 (53.3;73.0)	
Male gender	256 (65.0%)	112 (60.2%)	144 (69.2%)	0.061
DeBakey classification,				0.108
DeBakey I	292 (78.5%)	131 (74.9%)	161 (81.7%)	
DeBakey II	80 (21.5%)	44 (25.1%)	36 (18.3%)	
Logistic EuroSCORE I	28.7 (18.1; 43.6)	24.6 (16.1; 39.7)	31.8 (18.7; 47.9)	0.018
EuroSCORE II	6.6 (3.8; 13.3)	5.5 (3.6; 10.3)	7.5 (4.0; 15.9)	<0.001
Body mass index [kg/m^2^]	26.2 (23.9; 29.3)	26.3 (24.0; 29.4)	26.1 (23.8; 29.2)	0.933
Body mass index > 30 [kg/m^2^]	79 (20.1%)	36 (19.4%)	43 (20.8%)	0.726
Arterial hypertension	263 (66.8%)	125 (67.2%)	138 (66.3%)	0.857
Pulmonary hypertension	7 (1.8%)	4 (2.2%)	3 (1.4%)	0.712
Type 2 Diabetes mellitus	20 (5.1%)	7 (3.8%)	13 (6.3%)	0.262
Insulin dependent	6 (1.5%)	3 (1.6%)	3 (1.4%)	1.000
Hyperlipoproteinaemia	42 (10.7%)	22 (11.8%)	20 (9.6%)	0.477
Creatinine at admission > 200 [µmol/L]	17 (4.6%)	6 (3.4%)	11 (5.8%)	0.270
Chronic renal insufficiency	46 (11.7%)	16 (8.6%)	30 (14.4%)	0.072
Decompensated renal insufficiency	9 (2.3%)	3 (1.6%)	6 (2.9%)	0.510
Renal replacement therapy (“chron Dialyse”)	7 (1.8%)	4 (2.2%)	3 (1.4%)	0.712
COPD	26 (6.6%)	13 (7.0%)	13 (6.3%)	0.768
Peripheral vascular disease	15 (3.8%)	7 (3.8%)	8 (3.8%)	0.966
Smoking	75 (19.1%)	37 (19.9%)	38 (18.4%)	0.699
Coronary heart disease	68 (17.3%)	25 (13.4%)	43 (20.7%)	0.058
Heart rhythm				
Sinus rhythm	328 (83.2%)	159 (85.5%)	169 (81.3%)	0.261
Atrial fibrillation	54 (13.7%)	22 (11.8%)	32 (15.4%)	0.305
LVEF (%),	60 (55; 70)	60 (56; 70)	60 (55; 70)	0.200
Previous PCI	25 (6.4%)	6 (3.2%)	19 (9.2%)	0.016
Previous cardiac surgery	36 (9.1%)	21 (11.3%)	15 (7.2%)	0.161
Previous CABG	12 (3.0%)	5 (2.7%)	7 (3.4%)	0.696
IABP/ECLS	5 (1.3%)	0 (0.0%)	5 (2.4%)	0.063
Pericardial tamponade	71 (18.1%)	29 (15.6%)	42 (20.3%)	0.227
Marfan syndrome	11 (2.8%)	7 (3.8%)	4 (1.9%)	0.272
Bicuspid aortic valve	18 (4.7%)	8 (4.4%)	10 (4.9%)	0.849
Aortic valve vitium				0.987
Aortic valve stenosis	10 (2.6%)	5 (2.7%)	5 (2.5%)	1.000
Aortic valve insufficiency	133 (35.1%)	65 (35.7%)	68 (34.5%)	0.807
Combined Aortic valve vitium at Aortic valve replacement	6 (1.6%)	3 (1.6%)	3 (1.5%)	1.000
Neurological deficits	77 (19.5%)	22 (11.8%)	55 (26.4%)	<0.001
Clinical presentation				
Acute myocardial infarction (≤48 h)	14 (3.6%)	4 (2.2%)	10 (4.8%)	0.155
Cardiogenic shock	30 (7.6%)	10 (5.4%)	20 (9.7%)	0.110
CPR (≤48 h)	33 (8.4%)	3 (1.6%)	30 (14.4%)	<0.001
Transfer from intensive care unit	47 (11.9%)	16 (8.6%)	31 (14.9%)	0.054
Intubated at admission	44 (11.2%)	9 (4.8%)	35 (16.9%)	<0.001

**Table 2 jcm-10-05370-t002:** Operative data.

	All Patients (*n* = 394)	Combined Endpoint = 0 (*n* = 186/47.2%)	Combined Endpoint = 1 (*n* = 208/52.8%)	*p*-Value
Length of surgery [min]	275 (227; 340)	256 (218; 311)	288 (233; 358)	0.001
Cardiopulmonary bypass time [min]	167 (136; 212)	159 (130; 199)	180 (140; 228)	<0.001
Cross-clamp time [min]	92 (71; 132)	84 (65; 130)	96 (75; 134)	0.010
Circulatory arrest [min]	35 (26; 50)	32 (24; 42)	39 (28; 60)	<0.001
Number of packed red blood cells, unit	2.5 (0–16)	2 (0–16)	4 (0–16)	<0.001
Number of fresh frozen plasma, unit	0 (0–21)	0 (0–16)	1.5 (0–21)	0.031
Number of platelets, unit	2 (0–5)	2 (0–4)	2 (0–5)	0.002
Surgical procedure				
Single supracoronary replacement of the ascending aorta	187 (47.5%)	87 (46.8%)	100 (48.1%)	0.796
Partial arch replacement	94 (23.9%)	50 (27.0%)	44 (21.2%)	0.173
Total arch replacement	59 (15.0%)	15 (8.1%)	44 (21.2%)	<0.001
Conduit/Bentall operation	72 (18.3%)	35 (18.8%)	37 (17.8%)	0.792
David operation	24 (6.1%)	15 (8.1%)	9 (4.3%)	0.121
Elephant-trunk	9 (2.3%)	2 (1.1%)	7 (3.4%)	0.181
Associated with Aortic valve replacement	65 (16.5%)	30 (16.1%)	35 (16.8%)	0.852
Associated with CABG	29 (7.4%)	9 (4.8%)	20 (9.6%)	0.070
TEVAR(EVAR)	27 (6.9%)	10 (5.4%)	17 (8.2%)	0.267
Arterial cannulation				0.612
Femoral artery	62 (15.7%)	30 (16.1%)	32 (15.4%)	0.839
Ascending aorta	83 (21.1%)	33 (17.7%)	50 (24.0%)	0.126
Aortic arch	9 (2.3%)	4 (2.2%)	5 (2.4%)	1.000
Subclavian artery	1 (0.3%)	0 (0.0%)	1 (0.5%)	1.000
Apex	5 (1.3%)	2 (1.1%)	3 (1.4%)	1.000
Pulmonary vein	234 (59.4%)	117 (62.9%)	117 (56.3%)	0.179
Venous cannulation				
Right atrium	382 (97.2%)	183 (98.4%)	199 (96.1%)	0.328
Bicaval	3 (0.8%)	0 (0.0%)	3 (1.4%)	0.177
Femoral vein	8 (2.0%)	3 (1.6%)	5 (2.4%)	0.727

**Table 3 jcm-10-05370-t003:** Postoperative data and outcomes.

	All Patients (*n* = 394)	Combined Endpoint = 0 (*n* = 186/47.2%)	Combined Endpoint = 1 (*n* = 208/52.8%)	*p*-Value
48 h-drainage loss [mL]	900 (500; 1513)	750 (350; 1200)	1030 (650; 1878)	<0.001
Postoperative blood transfusion	290 (74.6%)	120 (64.5%)	170 (83.7%)	<0.001
Postoperative fresh frozen plasma	197 (50.6%)	75 (40.3%)	122 (60.1%)	<0.001
Postoperative platelets	183 (47.2%)	70 (37.6%)	113 (55.9%)	<0.001
24 h-Number of packed red blood cells, unit,	1 (0–17)	0 (0–17)	1 (0–15)	0.029
24 h-Number of fresh frozen plasma, unit,	0 (0–24)	0 (0–24)	0.5 (0–23)	<0.001
24 h-Number of platelets, unit,	0 (0–10)	0 (0–5)	0 (0–10)	<0.001
Total number of packed red blood cells, unit	4 (0–56)	2 (0–38)	6 (0–56)	<0.001
Total number of fresh frozen plasma, unit	1 (0–76)	0 (0–36)	4 (0–76)	<0.001
Total number of platelets, unit	0 (0–20)	0 (0–9)	1 (0–20)	<0.001
IABP/ECLS	11 (2.9%)	1 (0.5%)	10 (5.1%)	0.008
Reintubation	69 (17.5%)	11 (5.9%)	58 (27.9%)	<0.001
Tracheotomy	99 (25.1%)	0 (0.0%)	99 (47.6%)	<0.001
Re-admission to the ICU	36 (9.2%)	8 (4.3%)	28 (13.5%)	0.002
Postoperative delirium	72 (18.4%)	39 (21.1%)	33 (15.9%)	0.190
Postoperative myocardial infarction	6 (1.5%)	0 (0.0%)	6 (2.9%)	0.032
TIA/Stroke	94 (23.9%)	0 (0.0%)	94 (45.2%)	<0.001
CPR	27 (6.9%)	4 (2.2%)	23 (11.1%)	<0.001
Bronchopulmonary infection	58 (14.7%)	12 (6.5%)	46 (22.1%)	<0.001
Bacteriaemia/sepsis	19 (4.8%)	1 (0.5%)	18 (8.7%)	<0.001
Rethoracotomy	71 (18.0%)	15 (8.1%)	56 (26.9%)	<0.001
Sternal wound infection/VAC revision	6 (1.5%)	4 (2.2%)	2 (1.0%)	0.431
New –onset of Hemodialysis	85 (21.7%)	0 (0.0%)	85 (41.3%)	<0.001
Atrial fibrillation	41 (10.5%)	19 (10.2%)	22 (10.7%)	0.881
Ventilation time [h]	69 (20; 209)	24 (15; 57)	189 (81; 387)	<0.001
ICU time [d]	6 (2; 12)	4 (2; 6)	10 (4; 18)	<0.001
Postoperative days	10 (7; 19)	9 (7; 13)	13 (7; 23)	<0.001
7 d-Mortality	45 (11.4%)	0 (0.0%)	45 (21.6%)	<0.001
Hospital Mortality	64 (16.2%)	0 (0.0%)	64 (30.8%)	<0.001
Cardiac death	34 (53.1%)	0 (0.0%)	34 (53.1%)	-----
Cerebral death	6 (9.4%)	0 (0.0%)	6 (9.4%)	-----
Sepsis	2 (3.1%)	0 (0.0%)	2 (3.1%)	-----
MOF	22 (34.4%)	0 (0.0%)	22 (34.4%)	-----

**Table 4 jcm-10-05370-t004:** Multivariable analysis of risk factors for the combined endpoint.

Variable	*p*	Odds Ratio	Confidence Interval
Coronary heart disease	0.021	2.122	1.118–4.028
Neurological deficits	<0.001	3.598	1.985–6.521
CPR	0.001	8.993	2.501–32.343
Intubated at admission	0.033	2.512	1.077–5.861

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
