# Peer review of "Preoperative Predictors of Adverse Clinical Outcome in Emergent Repair of Acute Type A Aortic Dissection in 15 Year Follow Up"

_jcm, 2021, doi:10.3390/jcm10225370_

Round 1

Reviewer 1 Report

The authors aim to identify preoperative predictors of undesirable outcome in emergent repair of acute type A dissection - presenting their 15 year institutional experience with 94% completed follow up. The work off the data is quite impressive. 

It remains unclear (as data is not presented) if and how their change of strategy regarding atrial canulation did influence the outcome (i.e. neurological deficit) line 82-84. This should have been analyzed/presented in the data. 

Over all it is not new that the state at which the patient arrives is crucial for its postoperative outcome. Previous studies pointed out that not only state of arrival (preoperative factors Eursoscore etc.) are essential for estimating perioperative risk but also x-clamp, massive transfusion and other intraopertive factors remain mid- and long-term ourtcome.

The discussion section of the manuscript does show a detailed and likewise holistic view and urges the need for a better risk evaluation before surgery in order to provide better long-term results (regarding QoL and social functioning). 

The question remains whether or not the knowledge of undesirable outcome would guide clinicians in their decision making regarding emergent surgery (line 53-55)? It is more assumable that it might influence perioperative  management! The authors point out (line 213-215) that this remains an ethical dilemma. 

Despite the impressive follow up and data, I dare to contradict the authors in their conclusion that surgeons should prioritize treatment only for patient with prognostic better outcome - first, as there is not sufficient data yet and second, especially in Germany / West Europe with high density of cardiac surgery and heart centers.  

Author Response

The reviewer addresses several important issues, that we like to comment on: 

First, the concern if and how our change of strategy regarding atrial cannulation may have influenced the outcome. Overall, technology and perioperative care have much changed and improved during the past two decades and several factors may contribute. But despite improved surgical technique and strategies, we have not seen significant changes in mortality. Since several strategies have been gradually changed rather than at a certain point, we are unfortunately unable to analyze and present this with our data. But the reviewer brings a good point and we addressed the issue in the discussion.

Second, the reviewer contradicts our initial conclusion that surgeons should prioritize treatment only for patient with prognostic better outcome. The reviewer substantiates this by raising the concern that  - first, there is not sufficient data yet and second, especially in Germany / West Europe there is a high density of cardiac surgery and heart centers. We have therefore adjusted our conclusion. However, with aging population and increasing pressure on healthcare systems, socioeconomic factors become more and more a concern and we feel that if unfavorable outcome can be predicted, this concern should be revealed to a family even before emergent surgery. The decision to consent to surgery or shy away from aggressive management may differ between cultures but always needs to be respected by healthcare providers. Dependence on multiple machines postoperatively may be considered acceptable by some individuals while it may not by others.

Reviewer 2 Report

The topic is interesting and the long follow-up period is particularly important. However, the paper required some explanations/corrections.

Comment 1 Abstract: ”Our study aims to identify preoperative  conditions predictive of undesirable outcome to help guide for whom surgery would not be recommended anymore „- Due to the dramatically high mortality without surgery, it seems that this statement is inadequate

Comment 2 Line 62: ”Variants with aortic intramural hematoma and intimal tears without hematoma as well as pene- trating atherosclerotic ulcers were included”- How many cases of this type were included in the analysis? Did the occurrence of such variants have no effect on prognosis? Perhaps it would be advisable to consider specific types of AAAD as prognostic risk factors.

Comment 3  Line 67: “transesophageal echocardiography was 67 routinely performed after induction of general anesthesia and endotracheal intubation in 68 the operating room to evaluate heart valves for need for concomitant procedures”- Whether TLE was performed in all patients, or was the assessment of the aortic valve by TTE insufficient

Comment 4 Discussion- line 245: The presented diagnostic and therapeutic organizational model of AAAD is very good because it reduces the time of diagnosis. Since time plays such an important role, it would be advisable to include this parameter in the results along with the analysis of whether the time from suspicion of AAAD to full diagnosis and to the start of surgery does not affect the postoperative prognosis.

Author Response

The reviewer addresses several important issues, that we like to comment on:

Comment 1 Abstract: ”Our study aims to identify preoperative  conditions predictive of undesirable outcome to help guide for whom surgery would not be recommended anymore„- Due to the dramatically high mortality without surgery, it seems that this statement is inadequate

We have rephrased the section according to the reviewer’s suggestion

Comment 2 Line 62: ”Variants with aortic intramural hematoma and intimal tears without hematoma as well as pene- trating atherosclerotic ulcers were included”- How many cases of this type were included in the analysis? Did the occurrence of such variants have no effect on prognosis? Perhaps it would be advisable to consider specific types of AAAD as prognostic risk factors.

This is an interesting thought, unfortunately we do not have data specific enough to analyze subgroups.

Comment 3  Line 67: “transesophageal echocardiography was routinely performed after induction of general anesthesia and endotracheal intubation in the operating room to evaluate heart valves for need for concomitant procedures”- Whether TEE was performed in all patients, or was the assessment of the aortic valve by TTE insufficient

In our department acute type A-dissection is routinely diagnosed by orienting ultrasound examination followed by computer tomography, and thus structured TTE before surgery is not routinely performed. Therefore we use TEE in the operating theatre for structured echocardiographic examination and planning of the procedure.

Comment 4 Discussion- line 245: The presented diagnostic and therapeutic organizational model of AAAD is very good because it reduces the time of diagnosis. Since time plays such an important role, it would be advisable to include this parameter in the results along with the analysis of whether the time from suspicion of AAAD to full diagnosis and to the start of surgery does not affect the postoperative prognosis.

This is also an interesting thought, unfortunately we do not have the corresponding data to analyze subgroups.

Round 2

Reviewer 2 Report

I have no further comment